# The Strange Case of the Gallo-Italic Dialects of Sicily: Preservation and Innovation in Contact-Induced Change

**Alessandro De Angelis**

Department of Civiltà Antiche e Moderne (DICAM), University of Messina, 98122 Messina, Italy; adeangelis@unime.it

**Abstract:** The Gallo-Italic dialects widespread within central–eastern Sicily represent the result of the medieval immigration of settlers from southern Piedmont and Liguria, after the Norman conquest of the island (1061–1091). As far as the language spoken by these communities is concerned, an oddity arises: most of their lexical and syntactic features developed further through contact with neighboring varieties (such as, most notably, Sicilian), whereas, at a phonetic/phonological level, they have remained very conservative, largely maintaining their original northern characteristics. In the present paper, the possible causes underlying such a split are discussed: if the transfer of syntactic structures can be triggered by the presence of bilingual speakers who become progressively dominant (that is, more proficient) in Sicilian as L2, the preservation of the main phonetic/phonological features can represent a tool employed to the ends of emphasizing the identity of these new settlers from both an ethnic and linguistic perspective.

**Keywords:** contact-induced change; sociolinguistic dominance; resistance principle; Gallo-Italic; Italo-Romance southern dialects

## 1. The Gallo-Italic Varieties of Sicily

Gallo-Italic varieties spoken in central–eastern Sicily are the result of medieval immigration of settlers coming from southern Piedmont and Liguria (Petracco Sicardi 1969; Pfister 1988; S. C. Trovato 1998, 2013), after the Norman conquest of the island (1061–1091). This wave of immigration, which concluded in the second half of the 12th century, was motivated by the necessity to fill a demographic vacuum caused by the Norman war of occupation of the Sicilian territories. It was fostered by the marriages and subsequent union between the Altavilla lineage and the Aleramici family of Monferrato, which had the aim of suppressing any possible revolts by those Arab communities that remained on the island, especially in the eastern–southern areas.

Nowadays, Gallo-Italic is spoken in about 10 villages (S. C. Trovato 1998, p. 537; S. C. Trovato 2013 and see the maps at pp. 278–79), alongside the local variety of Sicilian[1]. However, in many other localities, especially in the provinces of Messina, Catania, and Enna, in which Sicilian dialects are spoken, Gallo-Italic features are still clearly detectable (S. C. Trovato 1998, pp. 538–39).

A particular aspect of such an immigration—which had a notable impact at a sociolinguistic level—concerns the typology of the new settlements. Namely, these northern colonies did not build new settlements; rather, they established themselves in villages or towns already inhabited by Sicilian people. If this situation produced rivalry and sometimes conflicts between the two different ethnic and linguistic communities, such a state of affairs resulted in the formation of bilingual speakers with differing degrees of proficiency in Sicilian as L2, depending on the different villages or towns where Gallo-Italic is or was spoken. For example, regarding the village of San Fratello, Tropea's (1974, p. 371) sociolinguistic analysis documents, «[ . . . ] una gagliarda vitalità e una granitica compattezza» (a lively vitality and a granitic compactness) of the Gallo-Italic dialect at the end of 1960s, whereas

Sicilian was spoken in a state of diglossia and restricted to formal or "high" contexts (see Section 3.1). A different picture emerges in other localities, such as Nicosia, Aidone, and Novara, in which Tropea (1966) documents a "symbiosis" with the Sicilian varieties over the same period, although with a striking difference between the three localities. In his sociolinguistic sketch of Nicosiano, Tropea (1966) attests to a positive attitude amongst its speakers towards their native language, due to the prestige enjoyed by the town, itself characterized by an ancient nobility. On the other hand, the Gallo-Italic vernacular of Aidone, a village of low prestige, was confined to informal and limited domains (in that it was spoken with friends or family, or by farmers at the workplace), and associated with a feeling of shame (hence the reluctance to speak Aidonese in public contexts). Lastly, Novarese enjoyed a good degree of prestige, even if with differences dependent on some variables, with low social classes, women, and the elderly preserving the ancient dialect more than men.

Nowadays, in the more or less 10 localities in which Gallo-Italic is still spoken, the sociolinguistic status of the language has changed—despite being positively perceived by its speakers in the past—due to the impact exerted by both the local variety of Sicilian and, especially, by that of regional Italian.

As for Sanfratellano, Foti (2013) notes a progressive attrition of Gallo-Italic and a growing speaker proficiency in regional Italian. Nowadays, Italian is spoken by the whole community, not only in formal contexts, but also in those informal domains before exclusively reserved for Gallo-Italic (Foti 2013, p. 22). However, the local dialect is still being kept alive. A 1997 inquiry on the language revealed an active competence of Gallo-Italic among local preschool, primary, and middle school students. A questionnaire carried out by Alfonzetti et al. (2000) showed that Gallo-Italic is still used. Even those speakers who continue to use Italian in their everyday life were reported as still having a passive competence of the local vernacular.

Given this, Gallo-Italic can be classified as an endangered language with a quite good level of vitality, at least in the villages of Nicosia and San Fratello, where it is spoken alongside both the local variety of Sicilian and regional Italian in a situation of trilingualism. In this context, it shows the typical phenomena relating to language attrition, due to its long-term exposure to Sicilian and, later, to regional Italian.

In the contact-induced change which took place in these varieties, no striking anomalies have been noted as of yet. At least through an analysis of both the lexicon and syntax (as well as of derivational morphology), Gallo-Italic varieties of Sicily seem to be located at the highest levels of Thomason and Kaufman's (1988, pp. 74–76) borrowing hierarchy, showing a considerable number of structural elements transferred by the donor languages, especially from Sicilian. Conversely, the phonetic system looks very conservative if compared with the other levels of analysis.

The main aim of this paper is that of accounting for this contrast, or rather clarifying why, on the one hand, phonetics (and phonology) are well preserved, whereas, on the other hand, the lexicon and syntax show a large amount of borrowings.

The paper is organized as follows: Section 2 is devoted to illustrating the innovations found both in the lexicon and in syntactic structures, vis-à-vis the preservation of the phonetic/phonological level, which shows typical northern characters; Section 3 documents the issues that the permeability of contact of some areas of the grammar vs. the resistance of other areas pose for the general theory of linguistic contact; Section 3.1 sets out Van Coetsem's framework, especially regarding the notion of dominance and its applicability to our case study; Section 3.2 illustrates the sociolinguistic motivations which lie at the basis of the preservation of the Gallo-Italic phonetic/phonological system; lastly, Section 4 contains some final remarks.

## 2. A Linguistic Sketch of Gallo-Italic: Conservative and Innovative Features

Although studies showing the percentage of inherited lexicon in the Gallo-Italic varieties are scarce, scholars agree that most of the vocabulary of these Gallo-Italic dialects

derives from contact with Sicilian (S. C. Trovato 1981, 2018; Foti 2013, p. 104)[2]. Indeed, most of the Gallo-Italic lexicon represents the result of a koineization due to progressive exposure in particular to the neighboring Sicilian dialects and (later) to regional Italian, with phenomena of mixing, leveling, and simplification, despite the general phonetic camouflage following that of the northern Italo-Romance type.

As far as morphology is concerned, some areas appear to be very conservative. S. C. Trovato (1998) quotes, among others, the following northern morphological characters: the outcome of the infinitive ending -ĀRE of the first conjugation as -*é*, a very early phenomenon documented in northwestern Italy (XII c.); the outcome of the verbal adjective and nominal ending -*ato* into -*á* and -*uto* into -*ú*. Furthermore, some Gallo-Italic localities display the Pres. Ind. first pl. ending -V+*ma* (Novarese -*emu*, the typical Ligurian ending; Nicosiano -*ema*, supposedly a neutralization between Lig. -*emu* and Piedm. -*uma*, attested in the 'Ligurised' area of southern Piedmont (Valle Scrivia and Val Borbera)). On the contrary, the derivational morphology of these varieties displays a massive influence from Sicilian.

The pervasiveness of this contact also clearly arises within the syntax. As far as the lack of those notable syntactic characteristics that are very common in contemporary northern Italo-Romance varieties is concerned, this is not very significant, inasmuch as it could depend on the period to which the immigration dates back. One example of this is that of the lack of clitic subjects, nowadays a notable distinguishing trait of the northern areas from which these settlers presumably migrated (see, e.g., Piedm. [(ti) ət.CL.SUBJ ˈmanʤes] 'You eat', Regis and Rivoira Forthcoming). Namely, in the medieval northern varieties, the subject pronoun deriving from the Latin nominative was free and could be stressed; cliticization only gradually appeared from the Renaissance period onwards (Parry 1993).

As far as innovations are concerned, Gallo-Italic varieties syntactically look very similar to the neighboring Sicilian dialects. Hereafter, we provide a non systematic account of some the more salient syntactic features documented in Gallo-Italic varieties of Sicily, which are shared by Sicilian dialects (as well as, in many cases, by southern Italo-Romance varieties as a whole), with the goal of demonstrating the broad Sicilianization of the Gallo-Italic syntactic structures.

*2.1. Southern Syntactic Features*

2.1.1. Nominal Phrase

Gallo-Italic varieties display the so-called "prepositional accusative", a strategy of Differential Object Marking (=DOM) very widespread in central–southern dialects as a whole (De Angelis 2022), that is the use of a prepositional marker (Sanfrat. *a*, Sperlinghese and Aidonese *da*) for encoding some direct objects (especially [+animate], [+human] referents, according to a well-known animacy hierarchy). An example is Sanfratellano, which consistently shows the prepositional accusative with nouns at the higher degrees of the animacy scale, that is with first, second, and third personal pronouns, proper nouns, and kinship terms, especially when the latter are headed by possessive adjectives (Militana 2017):

(1)      [tə        ˈvitːʃ        a              tu]
         you      see.PST.1SG   DOM          you
         'I (just) saw you!'

(2)      [atʃaˈmɛi̯    a              ˈʤwæni]
         call.PST.1SG   DOM           John
         'I called John'

(3)      [ˈvitːʃ       a              ta            ˈsuɔr]
         see.PST.1SG   DOM           your          sister
         'I saw your sister'

Gallo-Italic varieties display the 'want + past participle' periphrasis with a volitional meaning, attested in Sicilian dialects and in the regional Italian of the island, cf. e.g., reg.

It. *vuole lavata la macchina* 'He wants the car washed', as well as in southern dialects as a whole (Ledgeway 2016, p. 267). The construct has been examined in Nicosiano (Menza 2017); see, e.g.,

(4)  *vuò*         *strengiud'*         *a*         *man*
     want.3SG      held.PAST.PART.F     the         hand.F
     '(S)he wants his/her hand held'

The past participle agrees with the subject of the subordinate clause (in the example, *a man*) and it is codified only by verbs which can license the role of beneficiary (in the example, the unexpressed Subject).

### 2.1.2. Verbal Phrase

Gallo-Italic dialects display the deontic periphrasis 'to have' + Prep. + Infinitive. The use of this construct (also) for conveying the deontic meaning ('to have to') is widespread in Sicilian dialects (Amenta 2010; Di Caro 2019). In Nicosiano, this periphrasis is realized in many different forms: notice, e.g., the reduplication of the preposition both before (*d-*) and after (*a, da*) the verb in examples (6) and (7)[3]:

(5)  *amö*                 *da*              *partö*
     have.1PL             PREP              leave.INF
     'We have to leave'

(6)  *d-am'a*                              *sparagnè*
     PREP=have.1PL=PREP                    save.INF
     'We have to save'

(7)  *le*             *d-avì*           *da*           *fè*              *söutè*
     them            PREP=have.2pl      PREP           to do.INF        blow up.INF
     'You have to blow them up'

According to Menza (2017, 2019), the different placements of the preposition can be traceable to two main schemas: (a) 'to have' *da*; (b) *d-*'to have'. The analytic verbal phrase 'have' + *da* could be derived from the similar construction (*aviri da*), attested in some neighboring dialects (Enna and Gagliano Castelferrato, see Menza 2019, p. 66). The structure *d-* + 'to have' + *a/da* (ex. 6–7) can be analyzed as the result of a phenomenon of the (clitic) doubling of the preposition, which can also be placed before the verbal form.

### 2.1.3. Sentential Domain

As is well known, southern Italo-Romance dialects exhibit clitic climbing in restructuring contexts. In contrast both to Standard It., where climbing is optional (*ti voglio parlare* vs. *voglio parlarti* 'I want to speak to you') and some north(western) Italo-Romance dialects such as Piedmontese, where clitic climbing lacks (Manzini and Savoia 2005, III, p. 335), southern dialects, including Sicilian varieties, generally display obligatory clitic climbing on the main verb. The behavior of the Gallo-Italic varieties is similar in this respect. See, e.g., the following example from Aidonese (A. Trovato 2020):

(8)  *Gianni t'à*       *da*           *parrere*       *de*          *cucina*
     John             you=have.3SG    PREP            speak.INF      about cooking
     'John has to speak to you about cooking'

A possible sentence with the pronoun encliticized on the embedded verb is deemed ungrammatical:

(8.a)  *\*Gianni*        *ara*              *parrergghje*       *de cucina*
       John             have.3SG=PREP      speak.INF=him       about cooking
       'John has to speak to him about cooking'

Another contrast with St. It. and northern dialects concerns the behavior of clitics with the negative imperative. In this case, St. It. again displays the double option of clitic placement: *Non ber=la/non la=bere* 'Don't drink it!'. Kayne (2000) interprets the placement of the pronoun before the infinitive (*non la bere*) as evidence in favor of clitic climbing: *la* climbs on a modal verb superficially unexpressed. Aidonese, which largely parallels the

rest of the Gallo-Italic dialects of Sicily in displaying clitic climbing, licenses only the latter possibility, with the obligatory climbing of the clitic object *a* in (9):

| (9) | *Nan* | *t'* | *a* | *bìvire* |
|---|---|---|---|---|
| | NEG | you | it | drink.INF |
| | 'Don't drink it!' | | | |

Note that, in Aidonese, this structure semantically corresponds to a sentence with a deontic modal verb (*ài* in the example (10)), which seems to confirm Kayne's hypothesis regarding the climbing of the pronoun before the infinitive:

| (10) | *Nan te* | *l'ài* | *bivire* |
|---|---|---|---|
| | NEG you | it=have.2sg | drink.INF |
| | 'Don't drink it!' | | |

However, in the case of clitic climbing, a preservation of a feature of the Old northern Italo-Romance varieties cannot be ruled out, given that in these varieties clitic climbing was expected (see Parry 1995 for Old Piedmontese). In this case, the contact with Sicilian dialects may have reinforced a structural feature already present in the Gallo-Italic dialects arrived in Sicily.

So-called Pseudo-Coordination (=PseCo) is a structure in which a closed class of V1s (frequently a motion or an aspectual verb) is associated with a V2 by means of a linker, formally a coordinator, which in both Sicilian and Gallo-Italic dialects is represented by *a* (<AC) or by Ø (Giusti et al. 2022). The construction formally displays a coordinative structure in which both verbs are inflected in the same tense and mood and share the same grammatical person. Despite its formal shape, PseCo behaves as an actual subordinate construction by encoding a single, complex event and exhibiting some monoclausal properties, as revealed by the general ban on placing syntactic material between V1 and V2. With regard to Gallo-Italic dialects, we offer some data from San Fratello (in this variety, second sg. and third sg. persons show asyndetic PseCo (=marked as Ø in the ex. 11), see Militana 2019):

| (11) | [ˈvɛi̯ | ˈkjɛti | u | ˈpɛã] |
|---|---|---|---|---|
| | go.IMPV.2SG Ø | buy.IMPV.2SG | the | bread |
| | 'Go and buy the bread' | | | |

| (12) | [ˈvɛã | a | ˈfɛã | la | ˈʃpaza] |
|---|---|---|---|---|---|
| | go.PRS.IND.3PL | AND | do.PRS.IND.3PL | the | shopping |
| | 'They go grocery shopping' | | | | |

Some Gallo-Italic varieties in the province of Messina and in the northern part of the province of Catania display the so-called "loss of the infinitive" (Rohlfs [1965] 1972; Ledgeway 1998; Assenza 2008; De Angelis 2013, 2017). Due to prolonged contact with Italo-Greek, the Italo-Romance varieties spoken in central and southern Calabria, as well as in northeastern Sicily (and Salento), replace an infinitival embedded clause with a finite one headed by a complementizer, probably derived by Lat. MŎDO (or QUOMŎDO), which gives the outcome *mi* in Messinese[4]. Furthermore, they use the same subordinate pattern in obviative (that is, noncoreferential) structures instead of a finite subordinate clause headed by Sic./central-south. Cal. *chi* (It. *che*), with a significant difference between the two finite structures. Indeed, in (QUO)MŎDO-clauses (at least in most cases), the subordinate verb—as occurs in Italo-Greek—is invariably inflected in the present indicative, irrespective of the Romance *consecutio temporum*. See the following examples from the Gallo-Italic dialect of Montalbano Elicona (S. C. Trovato 2013, p. 286)[5]:

(13) [ˈelːo̞ ˈvwɔrȩ me ˈvaju]
he want.PRS.IND.3SG COMP go.PRS.IND.1SG
'He wants for me to go' (Lit. 'He wants that I go')

(14) [aȩ rːasˈʤone me te lːaˈmentȩ]
have.PRS.IND.2SG right COMP you complain.PRS.IND.2SG
'You're right to complain' (Lit. 'You're right that you complain')

As a result of Italo-Greek influence, the same pattern—both in northeastern Sicilian and in Gallo-Italic dialects of this area—is also used for encoding complementizer-headed main clauses with a volition meaning (Ammann and van der Auwera 2004; De Angelis 2017), frequently expressing maxims, proverbial sentences, formulaic wishes, or curses:

(15) [me s=aˈsːetːa]
COMP PRON.CL.REFL=sit down.3SG
'Sit down, please' (Montalbano Elicona (Messina), S. C. Trovato 2013, p. 286)

*2.2. Phonetic and Phonological System*

If compared with the noteworthy transfer of lexical items and syntactic patterns, the phonetic and phonological system of Gallo-Italic seems rather unaffected by this contact. The northwestern Italo-Romance 'look' that the phonetic and phonological system of the Gallo-Italic varieties of Sicily show is described in S. C. Trovato (1998, 2013).

2.2.1. Vocalism

Gallo-Italic varieties show a heptavocalic stressed vowel system, differing from the pentavocalic system found in the Sicilian dialects. This represents not only a striking contrast with neighboring Sicilian varieties, but it also contrasts with another case of contact that occurred in the same area. Namely, the Sicilian pentavocalic system probably represents the ultimate result of interference with the medieval Italo-Greek pentavocalic vowel system (Fanciullo 1984, 1996, pp. 17–22). Therefore, in this respect, the two languages behaved in the opposite way: the previous vowel system present in Sicilian varieties (presumably, a heptavocalic system) underwent change under the pressure exerted by Italo-Greek, whereas Gallo-Italic dialects preserved their original heptavocalic system, regardless of the pressure exerted by the Sicilian pentavocalic system. It is worth pointing out that the starting conditions which possibly triggered the interference process were the same in both cases; that is, the presence in both source languages (respectively, Sicilian and Italo-Greek) of a less rich system than that of the recipient language. In this case, reduction can be triggered by structural factors: «la suppression d'une distinction phonologique est plus apte à s'imposer aux parlers qui la possèdent qu'une distinction supplémentaire à s'introduire là où elle manque [the suppression of a phonological distinction is more likely to be imposed on speakers who already possess it, than a supplementary distinction can be introduced where it is lacking]» (Jakobson [1938] 2002, p. 32).

The diphthongization of Ĕ, Ŏ occurs in open syllables, sometimes in closed syllables, and before palatal consonants; see, e.g., Sanfrat. [ˈdːzjɛu] 'frost' < GĔLU, [ˈwɔli] 'oil' < ŎLEU (open syllable); [aˈnjɛu] 'ring' < ANĔLLU, [ˈnwɔtː] 'night' < NŎCTE (closed syllable) (Foti 2013, pp. 27–29). S. C. Trovato (1998, p. 543) considers this feature to be the most distinguishing characteristic in determining the Gallo-Italic origin of these varieties. In some localities, such as Piazza Armerina and Aidone, diphthongs became monophthongs. In this respect, the Gallo-Italic varieties of Sicily greatly differ from those neighboring Sicilian dialects with which they entered into contact. Namely, in these latter varieties, diphthongization (when it occurs) is triggered exclusively by metaphony, except in those localities where Gallo-Italic substratum left some traces (De Angelis 2014). Forms such as Sanfrat. [paˈrjeɖːa] 'pan', [ˈfrjɛva] 'fever'; Novar. [ˈpjẽrã] 'pain', [ˈfjetːsa] 'dregs', [ˈbjelːua] 'weasel', [ˈpjergua] 'pergola', with final -/a/, differ greatly from diphthongized forms found in northeastern Sicilian, where the diphthong is triggered exclusively by the final vowels -/i, u/.

In Sanfratellano, Ē and Ĭ merged into [aᵢ], nowadays [a] (and [ɔ]); see, e.g., Sanfrat. *[kaˈnai̯la] > [kaˈnala] 'candle' < CANDĒLA, [ˈnar] 'black' < NĬGRU; [pɔʃ:] 'fish' < PĬSCE ([Foti 2013](#), pp. 30–31). This vowel opening is documented in many other areas of northern Italo-Romance (see [Rohlfs 1966](#), par. 57, with examples of both outcomes, [a] and [ɔ], the latter in Valle Anzasca (Piedmont)). The same vowels merged into [ẹi] (later [ẹ]) in Nicosiano following an outcome widespread in northwestern Italo-Romance ([Rohlfs 1966](#), par. 55). The outcome [ẹ] is widespread in other Gallo-Italic localities as well, as in Montalbano, S. Piero Patti, Piazza Armerina, and Ferla (S. C. [Trovato 1998](#), p. 545).

The outcome [i] (< Ē, Ĭ), attested in some lexemes in the dialects of Sanfratello, Novara, Aidone and Nicosia—whilst matching the Sicilian outcome[6]—is paralleled by the same development attested in some northwestern Italo-Romance dialects (although limited to some words, see [Rohlfs 1966](#), par. 56); see, e.g., Sanfrat. [ˈfiɣar] 'liver' < *FĒCATU, Aidon. [buˈt:ija] 'shop' < APOTHĒCA, and Novar. [ˈvidua] 'widow' < VĬDUA.

The palatalization of stressed A produces different outcomes; see, e.g., Sanfrat. [ˈpæla] 'shovel' < PALA; [aˈnɛʃ:ər] 'to born' < NASCĔRE; [ˈsɛa̯rt] 'tailor' < SARTŌRE. It also involves all the infinitives of the first conjugation, e.g., Sanfrat. [at:ʃaˈmer] 'to call', Nicos. [nˈde] 'to go', etc. As far as northwestern Italo-Romance varieties are concerned, the palatalization in Piedmontese is morphologically constrained (it occurs only with first conjugation infinitives and with the -ĀRIU suffix, see [Regis and Rivoira Forthcoming](#)); whereas, in other northern varieties, such as the alpine dialects of Lombardy, it is lexically widespread ([Rohlfs 1966](#), par. 19).

In the dialects of Fantina and Bronte, stressed A produces a velar outcome; see, e.g., Fantina *kåtu* 'pail' and Bronte *bbånku* 'bench', which is similar to that which occurs in some dialects of southwestern Piedmont and in some areas of Liguria.

In Novarese (included the so-called "sinecìa", the group of rural villages close to Novara), U has become /y/ (only in elderly speakers); see, e.g., [ˈlymi] 'light, lamp' < LŪME; [my] 'mule' < MŪLU, the result of a highly documented process that is widespread in the northern Italo-Romance dialects.

2.2.2. Consonantism

Consonantism also highlights the notably northern look of these dialects. First of all, consonants in the internal position are generally degeminated, itself a typically northern Italo-Romance feature.

Lenition. Whilst not generalized to all of the Gallo-Italic varieties of Sicily, the sonorization of the original voiceless stops (with further developments), especially between vowels, clearly sets apart the Gallo-Italic varieties of Sicily from neighboring Sicilian dialects; see, e.g., Sanfrat. [saˈvar] 'to know' < *SAPĒRE, [ˈr:wɔra] 'wheel' < RŎTA, [aˈmiæɣa] 'friend' < AMĪCA ([Foti 2013](#)); Aidon. [furˈmia] 'ant' < FORMĪCA ([Pfister 2008](#), p. 13). The fricative -s- also undergoes lenition (> -[z]-) in all these dialects (as well as in the respective varieties of regional Italian).

Assibilation. C- and G- before /e, i/ produce—although unsystematically—an alveolar affricate (namely, the voiceless [ts] and the voiced [dz], respectively); see, e.g., Sanfrat. [ˈt:sɔnər] 'ash' < CĬNERE and [ˈd:zjɛu] 'bitter cold' < GĔLU. The voiced outcome is attested in the dialects of the mountain areas of Liguria and in some areas of southern Piedmont ([Foti 2013](#), p. 40). In intervocalic contexts, the outcome of -C+E/I- is generally -[ʒ]-, except in Novarese, which displays -[ʤ]-; see, e.g., Sanfrat. [ˈkrau̯ʒ] 'cross' vs. Novar. [ˈkruo̯d:ʒi] 'id.' < CRŬCE, whereas the outcome of -G+E/I- is -[dz]-; see, e.g., Sanfrat. [ˈtandzər] 'to paint' < TĬNGERE ([Foti 2013](#), p. 40). Furthermore, -SJ- produces the same outcome -[ʒ]-; see, e.g., Sanfrat. [baˈʒɛr] 'to kiss' < *BASJĀRE ([Foti 2013](#), p. 54).

Rhotacism of -L-. In some Gallo-Italic dialects, -L- produces -[ɾ]-. However, other varieties preserve -[l]-, which constitutes evidence in favor of a hypothesized medieval immigration of these latter colonies; see, e.g., Nicosiano [mọˈlin] 'mill' < MOLĪNU vs. other Gallo-Italic varieties (San Piero Patti, Montalbano, Bronte etc.) [muˈɾinu], [muˈɾinọ]. In northwestern varieties, rhotacism of -/l/-, attested in Ligurian as well as in southern

Piedmontese, is documented only since the XIII c. (see, e.g., *Anonimo Genovese*); therefore, those colonies that retained -/l/- preserve a northern feature that was present previous to the -/l/- > -/r/- change mentioned.

In contrast with all of these north(western) phonetic features, the most striking Sicilian phonetic characteristic that permeated into some of these Gallo-Italic dialects is the presence, especially in hypercorrected contexts, of retroflex sounds due to the change from -LL- to -[ɖ(ː)]- in the localities of San Fratello, Nicosia, Sperlinga, Piazza Armerina, and Aidone; see, e.g., Sanfrat. [ˈʃtɔɖa] 'star' < STĒLLA and [ˈʃpæɖa] 'shoulder' < SPATŬLA (Foti 2013, p. 42). Through a process of hypercorrection, the Sicilian intervocalic outcome -[ɖː]- historically spread into initial position; see, e.g., Sanfrat. [ˈɖːamant] 'lament' < LAMENTU and [ˈɖːeu̯na] 'moon' < LŪNA (Foti 2013, p. 41).

A possible question that arises from the phonetic outline detailed above is whether these changes represent productive processes or whether they are instead the final, lexical-ized outcomes of these processes. In the latter scenario, they no longer support the claim of conservativity of these varieties at the phonetic/phonological layer.

It is indeed possible that some of these features are no longer productive nowadays, but the crucial question for our proposal is that of whether they operated in the past (and, if they did, for how long). If we can prove that they were operating at least at some linguistic stage of the Gallo-Italic spoken in Sicily, then we can provide the evidence needed for supporting the proposal concerning the preservation of the original phonetic and phonological patterns. In this perspective, the lexicalization would constitute a later stage of development that possibly occurred once (and if) the productivity of these originally phonological processes came to an end.

The most striking evidence in favor of the original phonological character of these features comes from the adaptation of Sicilian (and Italian) loanwords. These clearly attest to the historical productivity of most of the abovementioned changes. Here, we list some examples, also including some changes not mentioned thus far[7].

Sicilian borrowings such as Sanfrat. [ˈnjɛʃː] 'I leave' < Messinese [ˈnɛʃːu], Fantina [ˈʃmjertʃʊ] 'peach' < Messin. [ˈzbɛrʤu] 'nectarine'[8], Montalbano [ˈlːwokːọ] 'here' < Messin. [ˈɖːɔku], and Sanfrat. [trəˈpːwɔːru], Nicos. [trəˈpwọdeno] 'tripod' < Sic. [ʈʈiˈpɔdu] show the (nonmetaphonetic) diphtongization (with subsequent outcomes), which affects the original middle–low vowels. S. C. Trovato (1998, p. 543) highlights that this diphthong is one of the first features to be dropped when speakers switch from their Gallo-Italic dialect to the local Sicilian variety. Notice that other (probable) Sicilian loanwords, such as Novar. [ʃkʊˈbːjetːa] 'rifle, carbine' < Sic. [skuˈpɛtːa]; [praˈɹjetːa] 'destiny, fate' < Sic. [praˈnɛta], and Nicos. [traˈtːsjɛra] < Sic. [ʈʈaˈtːsɛra] 'rural road' show diphthongization even in the presence of the final -/a/. This clearly suggests that this type of process has nothing to do with the central–eastern Sicilian (metaphonetic) diphthongization, the latter developing only if the final vowels are -/i, u/.

Borrowings such as Nicos. [ˈkɔpəla], [ˈkɔpọla] 'flat cap' < Sic. [ˈkɔpːula]; Nicos. [ˈskrɔpọ] 'stick, small piece of wood' < Sic. [ˈskrɔpːʊ]; Nicos. [suˈpjera], [sọˈpjera] 'soup dish' < Sic. [suˈpːɛra]; Nicos. [tːsaˈpọ̈] 'hoe' < Sic. [tːsaˈpːuni]; Nicos. [dọˈbɛ] 'settle for; get by, manage' < Sic. [adːuˈbːari]; Nicos. [ˈrːɔba] 'wealth, holdings' < Sic. [ˈrːɔbːa]; Nicos. [taˈbọna] 'portable stove' < Sic. [taˈbːuna]; Sanfrat. [taˈbutː], Nicos. [taˈbutọ] 'coffin' < Sic. [taˈbːutu]; Nicos. [ˈzọbọ] 'asphodel; stick type; penis' < Sic. [ˈdːzubːu]; Nicos. [ˈʃekọ] 'donkey' < Sic. [ˈʃɛkːu]; Nicos. [keciˈɛ] 'to babble' < Sic. [kicːiˈari]; Nicos. [travaˈɹɛ] 'to work' < Sic. [travaˈɹːari]; Nicos. [daˈmusọ], [dːaˈmusọ] 'traditional stone house' < Sic. [dːaˈmːusu]; Nicos. [kaˈnata] 'carafe' < Sic. [kaˈnːata]; Nicos. [ˈmɛtʃọ] 'wick' < Sic. [ˈmɛtːʃu] show the northern degemination of the double consonants between vowels. In Novarese, evidence of this feature dates back to a seventeenth-century manuscript, where reg. It. forms such as *obedienza* 'obedience' (It. *obbedienza*), *placa* 'badge' (It. *placca*), *propagina* ('tombstone', but lit. 'extremity', It. *propaggine*), etc., are attested (Abbamonte 2020, pp. 73–74). Had gemination been merely a lexicalized process, it is likely that forms such as those mentioned thus far would have been adapted to the phonological system of

Sicilian: for example, Sicilian varieties only allow /bː/ (and not /b/, as in Gallo-Italic) in their phonemic repertoire.

As far as lenition is concerned, Sicilian loanwords also attest to this change, and, subsequently, attest to the productivity of this development in the Gallo-Italic enclaves of Sicily; see, e.g., Novar. [lːazaɲːaðoe] 'rolling-pin' < Sic. [lasaɲːaˈturi]; Nicos. [mekaˈdo̯ro̯], [mo̯kaˈdo̯ro̯], Aidon. [məkaˈurə] 'handkerchief' < Sic. [mukːaˈturi]. Furthermore, forms such as Sanfrat. [ˈntaza] 'hearing' (litt. 'understanding') < Sic. [ˈntisa] and Novar. [kaˈo̯zʊ] 'boy' < Sic. [kaˈrusu] show the lenition of -[s]- into > -[z]. As in the case of degemination, if this process were to have ceased being productive, it is highly probable that Sicilian phonology would have changed -[z]- into -[s]- in loanwords.

Both processes, however, show counterexamples. Forms such as reg. It. (Nicosia) [avokaˈt:o̯] (cf. It. *a*[vː]*oca*[t]*o*) 'lawyer') and [ɔkomˈpratːo] 'I've bought' (cf. It. *ho* [kː]*ompra*[t]*o*) are hypercorrected forms that document a change presumably still in progress, aimed towards the acquisition of the [+length] feature in the consonantal system.

Further examples are *ballatta* 'tombstone; slab' (Nicos. and Sic. [baˈlata]); *pariotto* 'seasonal worker' (Nicos. [pariˈɔto] and Sic. [paˈrjɔtu]; *raccina* ([tːʃ]) 'grape' (Nicos. and Sic. [rːaˈtʃina]), with the intervocalic consonants realized as geminates in the reg. It. of the Nicosian lexicographer Antonino Campione (born in 1944, see La Rosa 2022).

Borrowings such as Sanfrat. [ˈtrupːa] 'troop' and [ˈkapːja] 'couple' with the preservation of the geminate consonant prove to be recent Italian loanwords.

Regarding lenition, counterexamples point to a series of different outputs aimed at preserving the original voiceless consonants of the L2. Beside the mere retention of the original voiceless stops, such as Sanfrat. [ˈjɛpa] 'bee', [kaˈpir] 'to understand', and [ˈsanak] 'mayor' (the latter two are recent loanwords), a different solution concerns the formation of non etymological geminated consonants, which, as such, take away potential contexts from the application of the lenition rule. See, e.g., Novarese pairs such as [brʊsˈkaðʊ] 'toasted', [maˈaðʊ] 'sick'—the genuine dialectal forms which show the lenition -T- > -[ð]—vs., respectively, [brʊstoˈlitːʊ] and [maˈatːʊ] (Abbamonte 2009–2010, p. LXI), which are hypercorrected loanwords. Similar forms are, e.g., Novar. [ˈlːypːʊ] 'wolf', [nəˈpːytɪ] 'nephew'; [aˈbe̯tːʊ] 'spruce', [vəˈtːəlːʊ] 'calf'; [ˈnakːa] 'cradle, crib', and [ˈnəvekːa] 'it is snowing'. Another differing output concerns the change of the original voiceless stops into geminated voiced ones; see, e.g., Novar. [ˈkabːʊ] 'boss, chief', [ˈkrabːa] 'goat', [sːaˈbːɛ] 'to know'; [ˈʃko̯bːa] 'broom', etc.

Note that forms in which lenition is lacking are not all to be considered recent loanwords nor even borrowings. For example, a form such as Novar. [ˈnakːa], which derives from Sic. [ˈnaka], an ancient Greek loanword, probably represents an early borrowing from Sicilian. Furthermore, at least some of the aforementioned Novarese forms with double consonants may pertain to the inherited lexicon.

Indeed, at least in some varieties such as Novarese, once intervocalic voiceless stops were adapted as double consonants in loanwords, these also spread through hypercorrection to the inherited lexicon, in contrast to that which occurred in other varieties, by changing the original system. Namely, in her survey of this dialect, Abbamonte (2015, pp. 241, 242, 244) states that /p, t, and k/ are generally realized as double between vowels (-[pː]-, -[tː]-, -[kː]-). On the other side, in Nicosiano, intervocalic voiceless stops are only attested in the loanwords (Trovato and Menza 2020, p. XIX), and the same is true for Sanfratellano.

Therefore, in those communities in which Gallo-Italic has been well preserved and has enjoyed a good level of prestige, this innovation has been very restricted (see Section 3.1). On the other hand, in Novarese, the spread of double consonants in the inherited lexicon also follows from sociolinguistic motivations. At the time of the Tropea's (1966) fieldwork, Novarese was still well preserved by the women and the older generations. In contrast, male speakers tended to abandon their native language after emigration to northern Italy for work and their subsequent return. In his fieldwork conducted for the *Atlante linguistico italiano* (ALI) in 1963 (Massobrio et al. 1995, p. 1013), Tropea stated that the dialect was strongly

"corrupted" by neighboring Romance varieties, such as urban Messinese. Regarding the village of Fondachelli—close to Novara—he stated that its inhabitants were embarrassed to speak in their native language. To conclude, the extensive penetration of alloglot traits as well as a feeling of shame towards their native language resulted in the acceptance of foreign features that were unfamiliar to Gallo-Italic phonetic/phonological system.

Some loanwords show assibilation, as, e.g., Sanfrat. [ˈkrjɛʒa] 'church' (also attested in other Gallo-Italic varieties), probably from Sic. [ˈkrɛsja], [ˈkrɛsa] (Morosi 1885, p. 416; Foti 2013, p. 48), which attests the Gallo-Italic change -SJ- > -[ʒ]- vs. the preservation of the original cluster in the Sic. outcome -[sj]-. Nicos. [tːseiˈrɔbesǫ] 'propolis' from Sic. [t͡ʃiˈrɔbːisu] document the assibilated outcome of C- before /e/.

Forms such as Nicos. [ˈsticǫ], Sanfrat. [ˈsticː], from Sic. [ˈsticːu] 'vulva' and Nicos. [bardǫ̃] 'packsaddle' from Sic. [varˈduni] show the Gallo-Italic weakening of final unstressed vowels (except -/a/).

We can conclude by quoting other developments peculiar to specific varieties. Tropea (1966) analytically documented the loss of -N-, -L-, and -R- in Novarese (with a later restoration in some forms), which also affects Sicilian loanwords; see, e.g., [krəstəˈjɛ̃ŏ] 'person' < Sic. [krisˈtjanu], [ˈprẽa] f. 'pregnant' < Sic. [ˈprɛna]; [naˈfia] 'a little' < Sic. [naˈfila]; [pərsɪˈɣaʊ] 'peach tree' < Mess. [pirsiˈkaru], [kastaˈɲɪːɛ] 'fig tree; chestnut wood' < Mess. [kastaˈɲːara]. Regarding the latter two forms, I must point out that, if we were dealing solely with a kind of morphological calquing involving the outcomes of the suffix -ARIUS, -ARIA—as an anonymous reviewer observes—we would expect, in the first form, the preservation of the original voiceless stop -[k]-. The change of this phoneme to -[ɣ]- shows the inability that Gallo-Italic speakers would have had in realizing voiceless stops in intervocalic position, which, consequently, shows the existence of a phonetic and phonological system different from that found in Sicilian varieties.

All of these facts confirm an active retention of the most salient northwestern phonetic and phonological features which the settlers brought into their varieties upon their arrival in Sicily. The acquisition of new features (such as allophonic consonant length in Novarese) are sporadic and limited to those Gallo-Italic dialects in which the prestige of Sicilian has been perceived as stronger. Note that, in Novarese, other changes also point towards the same phenomenon: the (sometimes irregular, that is, unetymological) restoration of -/n, r, and l/-, also in indigenous lexemes, are probably due to an attempt at "ennobling" the vernacular (Tropea 1966, p. 34).

## 3. Gallo-Italic and the Contact-Induced Change Theory

The conservative nature of the phonetic and phonological system, when compared with the massive influence of Sicilian found in both the lexicon and syntactic patterns, is rather surprising for at least three reasons. The first one lies in the fact that the adoption of new sounds, features, templates, or the phonematicization of phonetic distinctions present in the source language (that is, the donor language) is generally (with very few exceptions) the result of a massive borrowing of lexicon as the ultimate result of transfer of fabric (Klein et al. 2020, p. 75). Stated differently, the transfer of phonological features generally represents the result of transferring lexical material (Winford 2010, p. 175ff.). Even though a direct relationship between the amount of the borrowed lexicon and the possible changes in the phonetic and phonological inventories of the recipient language cannot be determined with any degree of certainty, it is most likely that such changes have a chance to occur especially when the amount of lexical borrowing is numerically significant. When we observe the huge amount of foreign lexicon borrowed in these Gallo-Italic dialects, we note that only a very limited number of foreign sounds entered the affected language (see Sections 2.2.1 and 2.2.2).

The second reason consists in one of the implications of Thomason and Kaufman's (1988) hierarchy: if a target language extensively accepts structural borrowing in an area of its grammar, for example, in its syntactic system (e.g., at level 4 or 5 of the hierarchy, as we can conceivably measure the level of interference exerted by the donor language(s) on

Gallo-Italic), it must also accept borrowing of other structural elements, including phonetic and phonemic features. In our case, the adoption of syntactic patterns from Sicilian would suggest a similar massive adoption of borrowing in other grammatical subsystems.

The third reason is one that is more debated. Structural interference, more specifically syntactic interference, is considered by some scholars a quite sporadic phenomenon. See the following quotations:

«Il arrive—en une mesure du reste assez faible et dans des situations très particulières— qu'on emprunte à une langue étrangère des petits mots à valeur grammaticale; on n'emprunte guère de vraies formes grammaticales. Ainsi l'on est toujours ramené à la même conclusion: ce qui s'emprunte, ce sont essentiellement des éléments du vocabulaire» (It can happen-although to a weak extent and under particular circumstances- that small grammatical words are borrowed from a foreign language; however, actual grammatical forms are not borrowed at all. In this way, one is led to the same conclusion: whatever is borrowed, it is especially represented by lexical words) (Meillet [1921] 1982, p. 87).

«Syntactic structure very rarely, if ever, gets borrowed» (Winford 2003, p. 97).

«[ . . . ] most historical linguists used to be reluctant to admit any contact explanation for a structural change» (Thomason 2010, p. 34).

«Whether or not syntax can be borrowed at all is still highly problematic. Although Thomason and Kaufman's view has proponents [ . . . ], many students of language contact are convinced that syntactic borrowing is impossible or nearly so [ . . . ]» (Sankoff 2001, p. 509).

This could depend on the fact that «syntax is in some way the "deepest" level of the grammar» (Thomason and Kaufman 1988, p. 118). Furthermore, we need to recall the inert nature of syntax; that is, its tendency towards conservativity in linguistic change (Keenan 2009). Although Thomason and Kaufman (1988) are opposed to this view, giving examples in favor of the fact that «syntactic interference is as common as phonological interference» (p. 118), and statements such as Winford's (2003) may be regarded as extreme, the fact remains that a clear divide exists within our case study between the maximum preservation of the phonetics/phonology, on the one hand, vs. the maximum innovation of the syntax, on the other hand. From a syntactic perspective, Gallo-Italic, in all respects, resembles a Sicilian variety (and, more generally, a southern dialect); from a phonetic and phonological point of view, it shows a frozen status through the preservation of some features dating back to the time of the original immigration.

*3.1. Van Coetsem's Framework*

The first question that we would like to answer concerns the adoption of syntactic Sicilian patterns in the Gallo-Italic varieties under scrutiny. Thomason and Kaufman (1988, p. 75) admit that moderate and structural borrowing through strong cultural pressure exerted by the donor language is indeed possible, whereby the native language is maintained but it can change through the contact with another language, later facing what they call a "typological disruption", i.e., a radical change which they place at the highest level (the fifth one) of their hierarchy. The second point of theoretical interest concerns the recipient language's aptitude in to accepting a massive structural transfer.

A relatively recent approach has been put forward by Van Coetsem (1988, 2000) which evaluates this adoption of structural features—in the context of contact-induced change— by the recipient language. In this model, the notion of "borrowing" is distinguished from the notion of "imposition". When the agent of the transfer is a speaker of the recipient language (=RL), we have borrowing (=RL agentivity). On the contrary, if the agent of the transfer is a speaker of the source language (=SL), we have imposition (=SL agentivity). At the core of van Coetsem's theory lies the psycholinguistic notion of linguistic dominance, a crucial factor independent from social dominance, the latter being due to the power or prestige exerted by a language: «A bilingual speaker [ . . . ] is linguistically dominant in the language in which he is most proficient and most fluent» (Van Coetsem 1995, p. 70). In the case of borrowing, speakers are dominant in the recipient language. On the contrary, in

the case of imposition, the dominant variety is the source language. Note that, under this approach, the same agents may use both kinds of agentivity in the same contact situation. This approach differs from Thomason and Kaufman's (1988) proposal, which distinguishes borrowing—that involves the maintenance of the L1 (with the adoption of foreign elements via linguistic contact)—from (substratum) interference, in which interference happens in the course of second-language acquisition, more specifically in a process of language shift, when imperfect learners of a L2 import elements of their native language into the language that they are in the process of acquiring.

Van Coetsem's framework involves an entirely different scenario (Winford 2005, p. 376ff.). Let us assume a situation in which L1 speakers adopt, by means of borrowing, foreign elements from a donor language with which they come into contact. Later, these same speakers acquire a growing proficiency in the L2 over time, from which they transfer patterns into their native language. The result is a combination of borrowing and imposition: by means of borrowing, the speakers import especially lexical material in their native language, in which they are dominant (in the van Coetsem sense). At the same time, when they become most proficient and most fluent in the source language, it is the latter that becomes the dominant language, from which bilingual speakers transfer structures into their (original) L1. During the process of imposition, transfer especially concerns phonology and grammatical features. In this perspective, bilingualism plays a key role, without which the adoption of structural features is rare or completely absent (whence the difficulty on the part of many scholars in admitting the transfer of structural patterns).

A notable example concerning a similar scenario is provided by the well-known case of Asia Minor Greek, which we will describe here following the two abovementioned approaches. In Thomason and Kaufman's (1988, pp. 93–95) model, the extensive adoption of Turkish features in the grammatical system of Cappadocian Greek represents a paradigmatic example of language maintenance with heavy structural borrowing (i.e., the fifth level of their hierarchy), in which structural elements are borrowed in all areas of grammar without any constraint and despite the typological differences between the two languages. In the application of the van Coetsem's model by Winford (2005, pp. 402–9), Asia Minor Greek represents, in contrast, an example of both borrowing and imposition. Namely, the influence of Turkish on phonology, noun morphology (new types of declension with agglutinative number suffixes), and derivational morphology, as well as on the syntax, suggests a level of agency on the part of Turkish-dominant bilinguals. This represents a case of imposition, rather than a case of mere borrowing. This scenario fits in well with the sociolinguistic situation of these Greek communities (Dawkins 1916; quoted in Winford 2005, p. 408). A growing proficiency in Turkish by Asia Minor Greek speakers was favored by the seasonal migrations of men to Constantinople, where they spoke Turkish with one another. Many women also used Turkish in the home, which later caused their children to grow up abandoning their native language. Note that, on a par with the Gallo-Italic communities investigated in this paper, Turks and Greeks often lived in the same village (Augustinos 1992; quoted in Winford 2005, p. 402), a situation which further favored the pervasiveness of contact.

Coming back to Gallo-Italic, the data from this variety is partially similar to that of Asia Minor Greek (except for the phonetic/phonological level); that is, a scenario characterized by a high degree of bilingualism (or trilingualism, see Foti 2013 for San Fratello and cf. Section 1), where the actors of the change are speakers which have proficiency both in the source and recipient language.

On a par with Cappadocian Greek, bilingualism (and later trilingualism, with the spread of regional Italian) must have also been a common phenomenon within the Gallo-Italic communities, and not just during recent times. As far as the centuries-old history of contact between Gallo-Italic and Sicilian is concerned, we are far more aware of recent situations of contact between these varieties, but know strikingly little about contact in the past. For example, with regard to Sanfratellano, the more recent sociolinguistic evidence at our disposal dates back to the end of the 1960s. According to Tropea's (1974) outline, this

community constituted a tight social network (in the sense of Milroy 1987) whose degree of closeness was triggered by a series of external factors: the relative geographic isolation of the village; the lack of immigration from the surrounding areas (which fostered many cases of endogamy); the low sociocultural standing of its inhabitants; a sort of diffidence and hostility towards the external environment. These factors, as a whole, alongside feelings of pride towards their native language, led to the local variety of Sicilian being very restricted in its usage, whereby it was limited to more formal registers or to interactions with foreign Sicilian speakers. Among other contexts, the use of Sicilian in families where there was a high school or a university graduate, with the consequent abandonment of the local variety, is very revealing of the prestige once exerted by this language, but, at the same time, is very indicative of substantial unfamiliarity of Sicilian within the Gallo-Italic group; in this context, «l'uso del siciliano era quasi una prerogativa o una sorta di privilegio» (the use of Sicilian was almost a prerogative or a sort of privilege) (Tropea 1974, p. 81).

However, between Tropea's sociolinguistic outline and Foti's (2013) recent survey that documents the spread of (regional) Italian and the replacement of Sicilian as the high variety in speaker repertoires (Foti 2013, p. 21 and see set. 1), there is a too short a time span for hypothesizing a potential extensive adoption of Sicilian elements into syntactic patterns of Gallo-Italic in the narrow space of around 40 years.

The only possible conclusion is that the Sanfratellano variety analyzed by Tropea represented a language already strongly influenced by Sicilian. In other words, over the centuries, the linguistic exchanges and relations between Sanfratellano and Sicilian must have been more intense and smoother, and the isolation of Sanfratellano at the end of the 1960s might not have been constant over time. Another piece of evidence on Sanfratellano from almost a century before Tropea's (1974) work identifies quite a different situation: «[ . . . ] per lo più il lombardo lo parlano come in Sanfratello i villani e i maestri e non i così detti galantuomini, i quali usano invece il siciliano comune» (for the most part, Gallo-Italic, as spoken in Sanfratello, is spoken by peasants and by teachers, but not by so-called gentlemen, who, on the contrary, use common Sicilian) (Pitrè 1872, p. 306; quoted in Trizzino 2020, p. 377, fn. 8). In contrast to Tropea's sociolinguistic survey, according to which Sicilian merely affected Gallo-Italic as a diaphasic variable and only in very limited circumstances, the Pitrè's description suggests a notably different scenario, in which the dominant variety seems to have played a larger role in the repertoire in causing a diastratically conditioned split.

Other communities, such as Nicosiano, would seem to attest to cases of ethnic and linguistic mixing. Nicosia was originally settled by followers of the Greek church, and later occupied by northern invaders after the Norman conquest of Sicily. The new inhabitants settled in the district of Santa Maria (so-called *Mariani*), whereas those indigenous followers of the Greek church established the district of San Nicolò (so-called *Nicoleti*). The local history of the village is full of bloody conflicts which point to such an ethnic contrast (S. C. Trovato 1997, p. 83ff.). However, the current dialect does not show traces of the past division, with few exceptions: in some areas of the district of San Nicolò, for example, /rː/ is realized as [rʲː]; /tr/ and /tːr/ are realized, respectively, as [tʑ] and [əː], just as with the local Sicilian outcomes. These developments are lacking in the (originally) Gallo-Italic districts of Santa Maria and San Michele (Trovato and Menza 2020, pp. XIII–XIV).

Another phonological change that occurred in the district of Santa Maria, that is, the development of /au/ > [ou] in unstressed syllables in prepausal contexts, provides evidence in favor of the spread of an original Gallo-Italic feature within the entire community. The analysis of such a feature in literary texts dating back to the first half of the 20th century shows its presence even in works belonging to authors from the district of San Nicolò. However, in the latter, it is found irrespective of the original contextual conditioning (Menza 2019, p. 60).

Moreover, the change from the rising diphthongs /je/ and /wo/ to the falling diphthongs, respectively [íe] and [úo]—nowadays attested in both districts—represents the spread of a feature which, in the past, was limited to the Gallo-Italic district of Santa

Maria, from which it was accepted by authors of the district of San Nicolò with a series of inconsistencies (Menza 2019, pp. 61–62).

Thus, phonetic microvariations point to an original sharp division between two different communities which disappeared almost completely over time in favor of the spread of Gallo-Italic features in all districts of the town (Trovato and Menza 2020, pp. XIII–XIV). By means of such a spread, contact and interference phenomena with the local Sicilian would have inevitable, including the adoption of syntactic features via imposition.

As we hypothesize for Sanfratellano, we claim that syntactic transfer did not take place in recent times, when Sicilian has been replaced by (regional) Italian as the prestige variety. Rather, many of the syntactic phenomena quoted in Sections 2.1.1–2.1.3 (as pseudo-coordination, Differential Object Marking, the construct 'want + past participle') are already attested in the poems of the Nicosiano writer Carmelo La Giglia (1862–1922) (Salvatore Menza, p.c.).

In Novarese, some syntactic patterns, such as Differential Object Marking, are also attested in the regional Italian—clearly influenced by the local Gallo-Italic vernacular, which is documented in a seventeenth-century manuscript (Abbamonte 2020).

*3.2. Phonetics as Last*

If bilingual speakers of Sicilian and Gallo-Italic—whilst becoming progressively dominant in Sicilian—imposed structural changes on their native language via SL agentivity, a question arises regarding the preservation of the main "northern" features at a phonetic (and phonological) level if compared with the massive adoption of syntactic constructions from Sicilian. If we recall the case of Asia minor Greek—a situation which seems partly comparable with the case study under investigation—we notice that structural changes via imposition affected all grammatical areas, including phonetics and phonology. It suffices to recall the Turkish vowel harmony extended to Greek inflections (Winford 2005, p. 404). In the case of Gallo-Italic, on the contrary, we are faced with a case of resistance towards change in a specific area of the grammar. The reason as to why some areas or subareas of grammar are impervious to external influence has been at the heart of the theoretical debate since at least Weinreich (1953, p. 44), with his mention of the "complex resistance to interference", in which not only structural reasons, but also «[ . . . ]socio-cultural factors (favorable or unfavorable prestige associations of the transferred or reproduced forms, etc.)» can play a decisive role in such preservation. Our hypothesis is that the strong conservation of the phonetic and phonological system, despite the large number of lexical items obtained via borrowing and structural features obtained via imposition, would hide any sociolinguistic motivations.

Phonetics (with the whole device of segmental and suprasegmental features) represents the most striking area of grammar regarding whose variation speakers have any awareness. Public attention towards accents has been given «from the Gileadites inability to realize palatal sibilants (*Old Testament*, *Judges* 12: 5–6)» (Preston and Niedzielski 2010, p. 2). Phonetics is the tool by means of which speakers of different linguistic communities identify each other, using empirical devices such as contrastive remarks, underlining the difference in pronunciation at prosodic level, or, on the contrary, by highlighting the possible similarities between their languages. The phonetic system is the breeding ground for the origin of sociophonetic markers by which speakers identify themselves as members of the community as opposed to the neighboring people (Labov 2001, pp. 215–21). Phonetic features can become markers of linguistic loyalty and they can create a real sense of pride and ownership within and towards a social network. On the contrary, they can constitute a tool for mockery and derision towards rival communities or single speakers.

It is likely that Gallo-Italic speakers—at least until regional Italian entered their linguistic repertoire to a more significant extent, see Section 1—preserved their original phonetic and phonological features as a tool for defending their identity, both from an ethnic and linguistic perspective, and for distinguishing themselves from surrounding Sicilian-speaking communities. Evidently, the need for emphasizing local features (and, consequently, local

identity) is greater the scarcer the regard or even contempt with which the Gallo-Italic people and all these dialects were dealt by Sicilian communities. Local traditions are full of popular and offensive nicknames towards Gallo-Italic communities and their language. Phrases as *parrèr a carcaràzza* ('to speak like a magpie'), *parrèr a v'ddanigna* ('to speak like a villain') in relation to the Aidonese Gallo-Italic dialect; the expression *câ fava ttâ bbucca* '(to speak) with a (fava) bean in one's mouth' referred to the inhabitants of Caltagirone, whose variety shows many Gallo-Italic characteristics (Cremona [1895] 2020); the nickname with which Sicilian people identifies the inhabitants of San Fratello (*menzalingua* 'half-tongue') clearly refers to the linguistic incomprehensibility of these varieties on the part of Sicilian speakers but to their offensive nature in general. Pairs such as *ciaccès ncaucà* ('the dialect of Piazza Armerina in its stricter form') vs. *chiazzìs c'ittadunu* ('the dialect of Piazza Armerina in its urban form') highlight the low sociolinguistic prestige enjoyed by the local Gallo-Italic variety. The epithet *zangrei* pl. 'rough, savage' with which the people of Sanfratello are dubbed—in relation to the nonindigenous nature of this community—clearly reveals the low consideration in which they are held (Trizzino 2021).

Under such conditions, this sort of phonetic resistance (regardless of the extensive disruption of the syntax in favor of Sicilian models) is aimed at preserving local identity, according to a well-known model in which local identity is proposed as motivating linguistic change (see at least Labov 2001).

## 4. Final Remarks

The present case study seems to confirm the validity of the model put forward by van Coetsem and applied by Winford (2005) to different cases of contact-induced changes, although with a caveat. In the case of Gallo-Italic, on a par with those situations of change by means of imposition detailed by Winford (2005), the presence of bilingual speakers seems to be the crucial condition for the adoption of massive structural features via SL agentivity. One might be surprised to observe the pervasiveness of contact in this area of the grammar, vis-à-vis the conservativity of the phonetic and phonological system. We posit this as being able to be explained by the prestige which speakers ascribe to both languages, i.e., the covert prestige towards their native language and the overt prestige towards the dominant language, respectively.

In this scenario, speakers of a heritage language are willing to give up entire portions of their grammar in favor of the dominant language patterns when they live in a condition of subalternity vis-à-vis the language in which they progressively acquire proficiency. However, even when they recognize the objective prestige of the dominant language, they can voluntarily maintain their local identity, especially in situations in which they feel threatened by a possible loss of ethnic and linguistic autonomy[9]. In such a situation—in which speakers are driven in their choices both by overt and covert prestige—they can make a clear split in the mechanisms through which the interference happens: they accept all structural transfer except in the linguistic level that they deem most suitable for the self-identification of an ethnic group; that is, except at the phonetic level. Thus, although these original northern varieties look highly influenced by Sicilian in their syntactic patterns, their speakers, due to the preservation of the original phonetic/phonological system, are still aware that they speak a heritage language clearly different and distinguishable from Sicilian: according to Vigo (1857, p. 48), speakers of Sanfratellano were aware that they speak a northern variety. Indeed, they claimed: *parduoma a dumbard* 'we speak lumbard' (*lombardo* originally 'inhabitant of northern Italy') in opposition to *parduoma a datin* litt. 'we speak Latin [=Sicilian]'.

**Funding:** This research received no external funding.

**Institutional Review Board Statement:** Not applicable.

**Informed Consent Statement:** Not applicable.

**Data Availability Statement:** Not applicable.

**Conflicts of Interest:** The author declares no conflict of interest.

## Notes

[1]  These villages are: San Fratello (with Acquedolci), San Piero Patti, Montalbano Elicona, Novara di Sicilia (with Fondachelli Fantina) (province of Messina); Randazzo (province of Catania); Nicosia, Sperlinga, Piazza Armerina, Aidone (province of Enna); Ferla (province of Siracusa).

[2]  S. C. Trovato (2018) collects just over 200 words noted as having a certain northern origin (that is, lexemes attested in those northern Italo-Romance areas from which Gallo-Italic settlers presumably migrated).

[3]  In Sicilian, the most widespread preposition which surfaces in this construction is *a*.

[4]  The final *-i* of *mi* could represent the crossing with Sic. *chi* (< QUĬD), see Rohlfs (1968, § 789).

[5]  The transcription in IPA represents an adjustment of the transcriptional system used in S. C. Trovato (2013). Vowel length is not represented.

[6]  See, e.g., Sic. /ˈtila/ 'cloth' < TĒLA; /ˈnivi/ 'snow' < NĬVE.

[7]  The data are mainly based on: Abbamonte (2009–2010), Abbamonte (2015); Foti (2013); Raccuglia (2003); Tropea (1966); S. C. Trovato (1981, 1998); Trovato and Menza (2020).

[8]  Rohlfs (1977, *sub voces*) documents Messinese area as having both the feminine form [ˈzbɛrʤa], [ˈzmɛrʤa] and the masculine form [ˈzmɛrʤu]. The Gallo-Italic form attested in Fantina presumably derives from the latter, with the application of the diphthongization, that in this variety happens both in open and closed syllable (S. C. Trovato 1998, p. 544).

[9]  For the sociolinguistic notion of "local identity" (or "territoriality"), see Labov (2001, p. 228) with references.

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
