# Peer review of "The Strange Case of the Gallo-Italic Dialects of Sicily: Preservation and Innovation in Contact-Induced Change"

_languages, doi:10.3390/languages8030163_

Round 1

Reviewer 1 Report

This paper develops an interesting thesis about these varieties, according to which their phonetic/phonological characteristics are very conservative, in striking contrast with their openness to morphosyntactic and lexical features from Sicilian. If this is correct, then the type of explanation offered is a plausible and important one. I say if it is correct because I have a fundamental reservation which needs to be addressed more carefully before the paper is ready for publication: that is that the alleged phonological phenomena may not really be phonological at all (let alone phonetic).

There seems to be some confusion in this article between diachrony and synchrony. The ‘phonological’ phenomena cited are indeed the product of historical phonological processes distinctive of northern Italo-Romance, but what is being discussed here seems to be, largely, the synchronically lexicalized results of these processes. To the extent that this is true, it is the lexicon not the phonology that is ‘conservative’, because its forms are those inherited from Gallo-Italic, rather than Sicilian ones. Only if we were discussing either active phonological processes, or items in the phonological inventory distinct from those of Sicilian, could we argue that we are talking about phonology. There is fact an example given of a distinctive inventory item (the vowel /y/ on p7) but it is implicit that this is actually recessive under Sicilian pressure. We are not given any examples of the ‘heptavocalic’ system, but this too might constitute an inventory difference, if the surrounding dialects absolutely do not have close vs mid vowel distinctions of any kind. To the extent that the results of the northern phonological processes (e.g., short consonants from long ones, diphthongs in closed syllables) are perfectly compatible with the Sicilian phonological inventory, we don’t have a real phonological difference. Overall, this study would be much more convincing if many more concrete examples from the phonology of both the Gallo-Italic and the surrounding Sicilian dialects were provided.

 It will be important for the author to have the text revised by some competent person who is a native speaker of English. At the moment there are places where the text might actually be unintelligible, or misleading, to a reader who didn’t also know Italian. I have simply not been able to mention all of these.

I add below some page-by-page comments on details

p1

‘Gallo-Italic’: I would suggest changing this passim to ‘Gallo-Italian’, which seems to me the more normal anglophone term.

‘Sicilian was spoken’ > ‘Sicilian is spoken’?

If the word ‘galliard’ exists in English, I don’t know it. ‘Lively’?

p2

‘On the other side’ > ‘On the other hand’

‘derivative’ > ‘derivational’

‘exposes the Van Coetsem’ > ‘sets out Van Coetsem’s’

‘exposure towards the substrate languages (e.g. Greek)’ > ‘exposure to…’. More seriously, is there any serious evidence of direct historical exposure to Greek?

‘characters’ > ‘characteristics’

‘Some areas of morphology too show themselves very conservative’ > ‘Some areas of morphology too show themselves to be very conservative.’ More examples could be given. If the morphology is conservative, the idea that ‘morphosyntax’ is more exposed to Sicilian influence than phonology is rather weakened.

p3

The gender of miele etc. is not a matter of morphology; if anything it is a matter of the lexical specification of gender (and hence of lexical conservatism). Overall, I think that the inflexional morphological differences between these dialects and Sicilian, especially in the verb (not just the 1pl), may be greater than is suggested here. At any rate we need more exemplification.

-é, a very ancient phenomenon in Northern 101 Italy (XII c.) More specifically, northwestern Italy.

‘totally lack’ > ‘are totally lacking’ (but in that case, how do we know?)

ˈsuɔ̯r (shouldn’t the subscript lunula be under the ‘u’?)

vuole lavata la macchina  Can’t this construction also be a passive: ‘La macchina va lavata’?

p5

‘As it is well-known’ > ‘As is well known’

‘we dispose of’; No! This would mean  ‘buttiamo via…’. Maybe ‘we possess’?

p6

facies This word isn’t used in English. Maybe ‘the northern appearance/look’?

‘an eptavocalic’ > ‘a heptavocalic’

p7

‘The fricative -s- too undergoes lenition (> -[z]-) in all these dialects’. I very much doubt it, although may have done so at some remote point in the past! On the other hand, one might argue that this constitutes a difference in phonological inventory between Sicilian and these dialects, since I don’t think that Sicilian has intervocalic [z]

‘entered’ ? ‘encountered?

‘due to the change of -L- into -[ɖ(ː)]-‘. More correctly (historically) ‘the change of -LL- into [ɖ(ː)]’. But what is most interesting here is that we apparently having a ‘Sicilianizing’ hypercorrection when the sound is introduced into initial position, and this rather goes against the general thesis of this article.

p8

‘the Thomason and Kaufman (1988)’s 381 hierarchy’ > ‘the Thomason and Kaufman (1988) hierarchy’

‘dans situations’ >  ‘dans des situations’

‘It turns out’ > ‘It can happen’; also ‘small grammatical words’

‘one leads’ > ‘one is led’

p9

‘by preserving even some features coming back to the epoque of the original immigration.’ > ‘by even preserving some features dating back to the time of the original immigration’.

The van Coetsem’s framework > Van Coetsem’s framework OR The Van Coetsem framework

p10

‘a time span too short ‘> ‘too short a time span’

p11

‘the Tropea’s sociolinguistic survey’ > ‘Tropea’s sociolinguistic survey’

‘cruent’ ?? ‘bloody’

‘/tr/ and /tːr/ are realized respectively as [tƨ] and [ƨː], according to the local Sicilian outcomes. These developments lack in the  (original) Gallo-Italic districts of Santa Maria and San Michele’ > ‘/tr/ and /tːr/ are realized respectively as [tƨ] and [ƨː], according to the local Sicilian outcomes. These developments are lacking in the (original) Gallo-Italic districts of Santa Maria and San Michele’.

By the way, this really does look like phonological conservatism, not in the sense of preserving local phonological features but this time in the sense of resisting encroaching Sicilian ones. this point needs to be made clearer.

‘change happened’ > ‘change that happened’

‘Santa Maria’s district’: ‘the district of Santa Maria’

‘the all’ > ‘all the’

‘San Nicolò’s district’: ‘the district of San Nicolò’

In what sense is the change mentioned here ‘an original Gallo-Italic feature’? We need more information.

‘become progressively dominant’ > ‘who had become progressively dominant’

p12

‘phonetics’: but are we really talking about the  ‘phonetics’ of Gallo-Italic in any way at all? It seems to me that what is mostly discussed in this study are the lexicalized results of old phonetic processes.

‘mokery’ > ‘mockery’

What are ‘blasons’??

‘into the mouth’ > ‘in one’s mouth’

Author Response

The reviewer states that "[...] what is being discussed here seems to be, largely, the synchronically lexicalized results of these processes. To the extent that this is true, it is the lexicon not the phonology that is ‘conservative’, because its forms are those inherited from Gallo-Italic, rather than Sicilian ones". 

In my opinion, this is not true. There is evidence that most of the phonetic features here described are (or they were, when Gallo-Italic settlers came in Sicily) ACTIVE processes. It is definitely possible that some of these features are not more productive nowadays, but the crucial question for our assumption is if they operated in the past (and, if they were, for how long). If we are able to prove that they were operating at least in some linguistic stage of the Gallo-Italic spoken in Sicily, then we can provide the evidence we need for supporting the assumption concerning the preservation of the original phonetic and phonological patterns. In this perspective, the lexicalization constitutes a later stage of development, occurred when (and if) the productivity of these originally phonological features ceased to act. The more striking evidence in favour of the originally phonological character of these features comes from the adaptation of Sicilian (and Italian) loanwords. These clearly attest the productivity (at least in the past) of most of the above mentioned changes. In the nwe version of my paper, I gave evidence in favour of such a productivity.

Reviewer 2 Report

This interesting article assesses the grammar of Gallo-Italic varieties in Sicily from the perspective of contact with Sicilian dialects and regional Italian. The authors point out that the different levels of grammar (phonological, lexical, morphosyntactic) are prone to contact phenomena/transfer to different degrees. While the morphosyntactic and lexical ones are largely permeated by Sicilian, the phonetic-phonological one is not: this is the ”oddity” they describe and aim to explain, with theoretical implications. They offer a sociolinguistic explanation for why this case-study is different from others such as that of Asia Minor Greek (where phonological phenomena are also transferred): as Gallo-Italic is sociolinguistically lower variety, it is associated with covert prestige, due to which speakers signal their identification with Gallo-Italic heritage/language through the phonology (which is particularly amenable for this purpose).

Some of the strengths of the article include its clear structuring/formulation of Sicilian Gallo-Italic grammar, in juxtaposing data presented in previous studies in such a way to capture its peculiarity and posing it as an open question that will be addressed. Following a useful and important socio-historical background, the study goals are very clearly laid out and followed throughout. Syntactic and phonological phenomena are well described, exemplified, contextualized.

The notion of “covert prestige” is used throughout the text, and is arguably central to the analysis, and my primary suggestion is that it could be more discussed/motivated. In particular in lines 622-5 and 630-1 it is suggested that Gallo-Italic phonological features are maintained due to covert prestige. But why would speakers want to assert/claim this identity if it was considered so lowly, as described in detail in the paragraph? Maybe the authors can elaborate more on the covert prestige which apparently drives them to do so. A final point is that it could be potentially useful to discuss van Coetsem’s model (and the notions of “borrowing” vs. “imposition”) in relation the notion of “heritage language” that is invoked in line 68 (in opposition to “endangered languages”).

Specific feedback:

Line 54: Gallo-Italic is said to be spoken in 10 localities, many of which are referenced throughout the article. It could be helpful to have these listed, perhaps in a footnote?

Lines 54-7, Here it seems unclear whether the change in Gallo-Italic status is in the positive or for the negative (if it “reduced its covert prestige”, then this implies that it is used less, even though towards the end of the paper it is precisely due to the vitality – rather than loss - of covert prestige that they conserve northern Italo-Romance phonetic-phonological characteristics, which is a bit confusing).

Line 68: Why juxtapose “heritage” and “endangered” language here? Perhaps a terminological clarification/periphrasis can be included.

Lines 108-116: It could be more directly stated that subject clitics are absent in the Gallo-Italic varieties of Sicily. As it is, it seems to be implied that since cliticization of subject clitics occurs only in the Renaissance period, they are not present in the Gallo-Italic varieties of Sicily, since the immigration occurred prior to this (If this is the observation, it could perhaps be stated more explicitly)

Line 161: The section on the deontic periphrasis could also mention that this is widespread in the southern Italian dialects in general / that it is frequently grammaticalized as a marker of future tense (Ledgeway, A., 2016, “The dialects of southern Italy”, in A. Ledgeway, M. Maiden (eds), 246-269); this text can also be referenced in the lines 145-7 on want passives).

Line 178: What is intended by “segment”?

Line 178-180: Can’t this be interpreted also as clitic doubling? “Da” doubling “a” below?

Line 207: It could be useful to recap what Kayne’s hypothesis that is being referenced here (and perhaps cited)

Line 212: "a control" structure implies it is biclausal; it would be useful to cite the main proponents of this view:

Manzini, Maria Rita, and Leonardo Savoia. 2005. I Dialetti Italiani e Romanci. Morfosintassi Generativa. Volume I. Alessandria: Edizioni dell’Orso. (cf. 688-501)

Manzini, Maria Rita, Paolo Lorusso, and Leonardo Savoia. 2017. “A/Bare Finite Complements in Southern Italian Varieties: Mono‐clausal or Bi‐clausal Syntax?” Quaderni Di Linguistica E Studi Orientali 3: 11–59.

Line 217-8: Unclear formulation

Line 231-32: Can also reference De Angelis’ monograph here: De Angelis, A., 2013, Strategie di complementazione frasale nell’estremo Meridione italiano, Messina, SGB.

Line 237-8: If the MODO + finite clause is distinguished in this explicit way from CHE + finite clause, then it could potentially be useful to mention how/if they are syntactically different (specifically in this non-coreferential context)

Line 267; 274-5: an eptavocalic à a heptavocalic

Line 275: It could be useful/disambiguating to add “under contact with Sicilian” when speaking of the Gallo-Italic retention of a heptavocalic system

Line 277-283: This elegantly described tendency towards simplification is confirmed in recent studies (though some exceptions are documented). A potentially relevant reference for this paper is: Andriani, L. & D'Alessandro, R. & Frasson, A. & van Osch, B. & Sorgini, L. & Terenghi, S., (2022) “Adding the microdimension to the study of language change in contact. Three case studies”, Glossa: a journal of general linguistics 7(1). doi: https://doi.org/10.16995/glossa.5748

Line 332: “voiceless consonants” à “voiceless stops/plosives”(?) (since the fricative /s/ is treated separately)

Line 360: I think this should be -LL- instead of -L-(?)

Line 362/3: “Due to the analogy with the Sicilian intervocalic outcome” perhaps as à “Through the hypercorrection of the Sicilian intervocalic outcome” (?) (“analogy with” can imply that it applies a distribution parallel/identical to the source variety, rather than a process of analogization to new contexts)

Line 370: “Replica”: Is this a technical term adopted in the literature? It could be useful if it was defined, is this referring to the "borrowing" language?

Line 293-4: I may be wrong, but there are two instances in which the French ‘a’ is perhaps à, as a should only be the lexical/auxiliary verb ‘to have’ which does not fit in these contexts (maybe the source text can be checked)

Line 430: Does “typological disruption” mean they no longer accept any changes?

Line 460-1: “From which bilingual speakers transfer structures into their ancestral language.” I may be not be following correctly, but shouldn’t this instead be into the L2?

Line 474: Does “noun” here refer to “lexis”?

Line 478-80: “of these Greek speakers” = native speakers of Greek? If I understand correctly, in this scenario, native speakers of Greek came to become more dominant in their (originally) L2 of Turkish(?) (If so, this could be explicitly mentioned), which is why we can speak of "imposition" into Greek rather than "borrowing"(?)

582-3: The function of the quote is unclear (“public attention on accents has been putted [put]” à “has been documented”(?)

587-592: Though referenced below, a citation of Labov’s sociophonetic studies could be useful here

Thomason and Kaufman (1988)’s hierarchy and relevant stages are mentioned multiple times throughout the text, it could be useful for this to be summarized/schematized at some point. Even though it is not the key model that is applied, it is used as an important starting point in the characterization of the Gallo-Italic “oddity”, and would in this sense be useful to be visualized more clearly.

Some terms that are used seem a bit unclear, and it would be useful to have described/briefly defined, as meaning is hindered: “maintenance” (I.e. line 447 and elsewhere), “ancestral language” (is this the L1? The native language?) “SL agentivity” (563, 620) - is this tied to concluding remarks  in 626 about speakers “[consciously?] willing to give up entire portions of their grammar”?

Examples/Glosses

Examples 1 & 3 the gloss for ‘saw’ à ‘see’

Inconsistent glossing of clitics (see examples 1, 7, as well as 8); in example 1, the case is given but not the person (which is the opposite of what is found in example 8), in example 7, only a translation is provided. This is also different from (8a & 9)

Example 10 is not glossed

Glossing of the present indicative is inconsistent (also in examples 4, 5-8a --> these include the person but not the mood) see also example 13-4

Example 12, a suggestion: ‘a’ (the clause linker) can maybe be glossed in small caps as a grammatical item (as and or ac or ad) since synchronically it is not a coordinator

Author Response

The reviewer asks why would speakers want to assert/claim their identity if it was considered so lowly. 

In my opinion, this depends on the sense of identity and ownership to the community. The need of underlying the local identity suffices to preserve (some of) the original features of a language, as showed by Labov in the case-study relating to Martha's Vineyard. In the pp. 14-15 of the new version of the manuscript, I underlined that "[...] the need for emphasizing the local features (and, consequently, the local identity) is greater the more is the scarce regard or even the contempt with which the Gallo-Italic dialects were dealt by Sicilian communities". 

Reviewer 3 Report

This paper is a welcome contribution to a rather understudied phenomenon regarding the sociolinguistic relationship among Sicilian varieties and the Gallo-Italic varieties spoken in 10 villages in Sicily. By mainly focusing on two documented cases from the dialects spoken in Sanfratello and Nicosia, the author(s) highlight(s) an unexpected outcome of the prolonged contact between these two groups of Romance languages: whereas Gallo-Italic varieties display a Sicilian behaviour as regards different syntactic structures, derivative morphology, and a high number of lexical elements, when it comes to phonetic/phonological aspects, they remain very conservative and quite distinct from Sicilian, keeping those phonetic/phonological features that characterised the original southern Piedmont and Liguria varieties whose speakers began to move to Sicily in 1061.

Since Thomason and Kaufman’s (1988) highest levels of borrowing hierarchy, the ones that are relevant in the case at study, would not explain for this oddity, the author(s) refer(s) to Van Coetsem’s (1998; 2000) framework and its psycholinguistic notion of language dominance, as used in Winford’s research. The linguistic scenario of Sicilian Gallo-Italic is thus crucially described as one of bilingualism (and then of trilingualism with the later addition of regional Italian), in which it is only when bilingual speakers become dominant in the source language (i.e. Sicilian) that they can ‘impose’ structural change to the Gallo-Italic varieties they speak. And it is in the same circumstances that they decide to preserve their original northern phonetic and phonological features, as a sociolinguistic way to mark their identity, in a region where the contempt towards their varieties by Sicilian speakers can be detected in a number of expressions underlining their lower language prestige.

This state of affairs implies a change in the way we conceive of the relationship between Sicilian and Gallo-Italic, which is why this paper represents an important contribution to the relevant topic: the idea of a continuous general isolation of the Gallo-Italic communities in Sicily must be discarded in favour of one that considers a long lasting linguistic contact with Sicilian, with sociolinguistic surveys accounting for the isolation of a given variety more likely to be exceptions limited to a specific period of time.   

This being said, the English used in this paper can be sometimes difficult to read by non-Italian readers and some paragraphs should be rephrased. Moreover, the paper looks rather sketchy in some sections, especially those regarding the syntactic structures discussed and the final remarks. One would expect the latter to contain also some ideas for further research about, at least, new sociolinguistic fieldwork among younger generations of bilingual speakers in the Gallo-Italic villages under discussion. In this sense, the work by Birdsong, Gertken and Amengual and their Bilingual Language Profile, offering a different view on the concept of language dominance, could be taken into consideration.

Nevertheless, I don’t think these facts impinge on the publishability of this contribution, that just needs some minor revisions. 

Here is a list of more detailed comments:

line 28-30: this claim could be supported by some references. 

line 50: footnote 4 should not be there.

lines 63-70: I’m afraid that in a linguistic scenario like that of Gallo-Italic communities in Sicily, the study by Alfonzetti, Assenza and Trovato (2000) is likely to be a bit too old-fashioned and not representative of the younger generations. The claim that Gallo-Italic cannot be classified as an endangered language, in 2023, might be reconsidered. 

lines 177-180: it seems to me that in (6) and (7) the preposition has not only undergone clitic climbing but rather reduplication, since a copy also occurs before the lexical verbs sparagnè (surfacing as a) and (as da).

line 185: there is a general inconsistency in the English renditions of the dialectal examples, which sometimes look more as literal or word by word translations (cf also (13), line 246) and sometimes as real translations.

line 207: the reference to Kayne’s work lacks the date. More importantly, it is never presented in the paper.

line 212: by labelling PseCo “a control structure”, the author(s) adopt(s) an account dating back to Manzini & Savoia (2005). However, they should recognise that a different account (dating back to Cardinaletti and Giusti 2001) that considers PseCo as a monoclausal construction is well represented in the relevant literature, too (both accounts are found in Giusti, Di Caro and Ross 2022).

line 559: “+ inf.” is probably a typo.

line 659: Amenta, Luisa >>> Amenta, Luisa. 2010.

line 687: Giusti, Giuliana, Vincenzo Di Caro, and Daniel Ross (eds.). 2022 >>> Giusti, Giuliana, Vincenzo Nicolò Di Caro, and Daniel Ross (eds.). 2022.

line 694: Klein, Thomas B., Anthony Grant, and E-Ching Ng >>> Klein, Thomas B, E-Ching Ng, and Anthony Grant

Additional references

Birdsong, D.; Gertken, L.M.; Amengual, M. (2012). Bilingual Language Profile An Easy-to-Use Instrument to Assess Bilingualism. COERLL, University of Texas at Austin. https://sites.la.utexas.edu/bilingual/

Cardinaletti, Anna & Giusti, Giuliana. 2001. “Semi-lexical” Motion Verbs in Romance and Germanic. In Semi-lexical categories, Norbert Corver & Henk Van Riemsdijk (eds), 371–414. Berlin: De Gruyter. https://doi.org/10.1515/9783110874006.371

Manzini, Rita & Savoia, Leonardo. 2005. I dialetti Italiani e Romanci. Morfosintassi Generativa, vol. I: Introduzione – Il soggetto – La struttura del complementatore, frasi interrogative, relative e aspetti della subordinazione. Alessandria: Edizioni dell’Orso.

Author Response

The reviewer observes, among other things, that "the English used in this paper can be sometimes difficult to read by non-Italian readers and some paragraphs should be rephrased. Moreover, the paper looks rather sketchy in some sections, especially those regarding the syntactic structures discussed and the final remarks".

In the new version of the manuscript, I improved especially the syntactic section.

Round 2

Reviewer 1 Report

In the time available I have only been able to look closely at the parts where the author seeks to answer my original objection that much of what is alleged to be the survival of an active Gallo-Italic 'phonetic system' is actually a matter of lexical survival of the results of those processes. I have to say that I remain unconvinced, and it does not seem to me that a compelling case for the central thesis of this paper has been made. Nonetheless, the author has attempted to address the objection so perhaps s/he deserves publication of the study so that others can assess the evidence.

 In particular, I had the following reservations, which the author should think about:

All the examples given of alleged Gallo-Italian diphthongization are also cases in which in Sicilian one would find metaphonic diphthongization, namely before (original) -u (notably ˈʃmjerʧʊ which cannot come directly from feminine ˈzmɛrʤa). These diphthongs may not be present in (modern?) Messinese, but this type of metaphonic diphthongization is amply attested, so far as I can see, in north-eastern Sicily.

The alleged examples of ‘degemination’ of Sicilian loanwords do not necessarily show that the northern process of degemination was still active. It simply shows that the Gallo-Italic dialects of Sicily did not originally have long consonants. There is a risk here of an anachronistic confusion of a difference in the phonological inventory with an active process which takes as its input geminates and degeminates them. The hypercorrections mentioned later are not evidence of any kind of ‘process’, but of accommodation to a difference in phonemic inventories.

The alleged examples of lenition strikingly nearly all concern derivational suffixes which presumably have equivalents in Gallo-Italic dialects. So is this active ‘lenition’, or is it a kind of morphological ‘calquing’? ‘ntaza’ is perhaps a more convincing case, but I would want to feel sure that this must be a loan from Sicilian and cannot be an inherited form. The same seems to me to be true of most of the examples alleged to illustrate the survival of the process deleting intervocalic -l-, -n-, -r-, mentioned later. In the case of foˈðau, by the way, there is no reason to believe that this is a loan from Sicilian: it is simply the normal inherited Gallo-Italic forms (see AIS map 1573). And fəˈɣaʊ cannot possibly come directly from fiˈkara; again I suspect it is simply the inherited Gallo-Italic form.

Finally, if this article is to be published there is a really urgent need for a thorough revision of the English by a competent native speaker. At the moment, there are passages which are likely not to be intelligible to readers who don't also know Italian! There is no reason why specialists in Italian dialectology should be expected to be good at writing academic English, of course, but that doesn't remove the need for someone to have a careful look at the text so that the thesis of the article can be clearly understood.

Author Response

I appreciated very much the time that the reviewer spent for reading my paper. Nevertheless, I disagree with him about the lexicalization of the phonological rules. In my opinion, if Sicilian forms undergo phonetic changes according the Gallo-Italic pattern (as they do, see e.g. the degemination of double consonants), then we can conclude that these changes represent the outcome of productive/active phonological rules. We don't know if all these rules are productive in the modern dialect, but we can state that these were operating in the past, as the Sicilian loanwords show (however, a "stratigraphic analysis" on the diachrony of Sicilian loanword in Gallo-Italic still lacks).

In some cases, we can doubt about the Sicilian origin of some words (it could be the case of foˈðau or fəˈɣaʊ indicated by the reviewer), but in many other cases there is no doubt about the Sicilian origin, see, e.g. Sanfrat. [ˈntaza] ‘hearing’ (litt. ‘understanding’) < Sic. [ˈntisa]; Novar. [kaˈo̝zʊ] ‘boy’ < Sic. [kaˈrusu]. As in the case of degemination, if this process were to have ceased being productive, it is highly probable that Sicilian phonology would have changed -[z]- into -[s]- in loanwords.  

Regarding the diphthongization, I added evidence supporting the fact that Gallo-Italic diphthongs have a different source from (north-eastern) Sicilian diphthongization, which is triggered only by final -/i, u/ (that is, it is a metaphonetic diphthongization). Forms such as Sanfrat. [paˈrjeɖːa] ‘pan’, [ˈfrjɛva] ‘fever’; Novar. [ˈpjẽrã] ‘pain’, [ˈfjetːsa] ‘dregs’, [ˈbjelːua] ‘weasel’, [ˈpjergua] ‘pergola’ show non metaphonetic diphthongization, as demonstrated by the presence of final -/a/, that in Sicilian varieties (as in all the metaphonetic Italo-Romance areas) never triggers diphthongization. 

The reviewer righthly notes that Gallo-It. [ˈʃmjerʧʊ] cannot come directly from Sic. feminine [ˈzmɛrʤa]. However, in north-eastern Sicilian also a masculin form [ˈzmɛrʤu] is documented (Rohlfs, SVS s.v.). The Gallo-Italic form attested in Fantina, [ˈʃmjerʧʊ], presumably derives from the latter, with the application of the Gallo-Italic (non metaphonetic) diphthongization, that in this variety happens both in open and closed syllable.    

Lastly, English has been revised by a native speaker. 

Reviewer 2 Report

This second draft applies many changes as a result of the first review stage, and I think this has resulted in a positive, strong outcome, with the thesis being articulated in a more clear and convincing way.

Following a revision of the English, I think this it is ready for publication.

A few notes here:

Line 183:  wouldn't it be more correct to talk about clitic 'doubling' since a copy of the preposition is conserved? I imagine the author disagrees with this as I briefly pointed this out in the last review.

Throughout: Where 'Italiot Greek' is written, it should be changed to 'Italo-Greek'

Line 432: the "change in progress" here is specifically degemination, correct? Since both lenition and degemination are spoken about in this passage, it would be useful to explicitly mention this throughout, i.e. which changes are presumed to be active or not.

Line 567: van Coetsem's model might be termed 'relatively recent' rather than "new" since it is over 20 years old. Also, it is unnecessary to use the definite article when discussing the model (i.e. "the van Coetsem's model") especially where this is not applied to Thomason & Kaufman's model (no article used), for consistency.

Lines 610-614: it would be useful for the author to use the key terms "borrowing" (612-3) and "imposition" (610) (?) where they apply, as it would facilitate the application of this theoretical model to the data, on the part of the reader

The same goes for lines 664-674 in the case of Gallo-Italic - Sicilian bilingualism. It is mentioned in the following sections (section 3.2) but I think since this is a central point of the thesis it would be very helpful, and more compelling for the argument, for this to be articulated clearly. This is the most important (light) modification that I think could be applied.

Author Response

I accepted all the changes that the reviewer suggested me.